# Study on Characteristics for Reaming Titanium Alloy Ti6Al4V with Two Kinds of Cemented-Carbide Groove Reamers

**DOI:** 10.3390/ma15145027

**Published:** 2022-07-19

**Authors:** Yongqiang Zhang, Yongguo Wang, Zhanlong Han

**Affiliations:** 1School of Mechatronics Engineering and Automation, Shanghai University, Shanghai 200444, China; yongqiang@shu.edu.cn; 2Guohong Tool System (Wuxi) Co., Ltd., Wuxi 214101, China; hanzl1208@163.com

**Keywords:** Ti6Al4V, reaming, groove structure, surface quality, geometric accuracy

## Abstract

Titanium alloys have been extensively used in practical machining owing to their outstanding mechanical properties, high specific strength and low thermal deformation. In this study, the cutting experiments are carried out on Ti6Al4V material with right-hand and straight cemented-carbide groove reamers. The experimental results show that the cutting force with the right-hand reamer is smaller compared to straight groove reamer due to the groove structure. The main tool wear forms are micro-chipping, adhesive wear, abrasive wear, and coating falling off on the right-hand reamer, while there is a built-up edge and serious damage failure on the cutting edge of the straight groove reamer. Notch wear and pitting on the surface of the hole wall are mainly caused by chip adhesion and tool wear. The surface-roughness value is the lowest as the cutting speed is 60 m/min and the feed rate is 0.4 mm/rev. The holes machined by the right-hand reamer have a low hole diameter deviation with various cutting parameters. The geometric accuracy of cylindricity is higher as the feed rate is 0.4 mm/rev and the cutting speed is 40 m/min for both kinds of reamers, and the cylindricity is better with the right-hand reamer.

## 1. Introduction

Titanium alloys have been broadly used in aerospace fields, chemical processing, biomedical science, as well as the automotive and nuclear industries due to their high specific strength, strength to weight ratio, fracture resistance, and excellent anti-corrosion properties [1,2]. However, titanium alloys are considered hard materials to machine because of their high temperature strength, high chemical activity, low elastic modulus, low thermal conductivity, and low deformation coefficient [3,4].

Reaming is usually used to machine precise cylindrical holes or conical cavities after drilling or boring, and it is a machining process to improve the geometric accuracy of holes [5]. The characteristics of reaming can be summarized as expanding the holes to the precision tolerance class and improving the surface finish to meet dimensional tolerance and surface quality requirements [6].

Various problems are encountered during machining. During the machining of titanium alloys, tool wear is an inevitable problem. Tool wear mechanisms usually include abrasion, adhesion, oxidation, and diffusion. Saketi et al. [7] investigated the pit wear and flank wear mechanisms of carbide inserts during the machining of Ti6Al4V, and found that different combinations of cutting parameters could lead to different wear mechanisms. Li et al. [8] studied the evolution of TiAlN-coated cemented-carbide tool wear during the drilling of Ti6Al4V, and found that the combined action of chipping, fracture, diffusion, and oxidation has an important influence on the failure of the front cutting edge, while the flank wear plays a dominant role on the outer cutting edges. Vijay et al. [9] investigated the tool wear and cutting force during machining with different coated tools. It was found that the crater and side wear rate of TiAlN-coated tools are smaller, and the cutting force is smaller, showing better machining performance.

The surface quality and geometric accuracy of holes are important indexes to evaluate the machining quality of reaming. Melo et al. [10] found that the spiral groove reamer has an important influence on surface roughness and cutting forces. In addition, choosing proper machining parameters has an important influence on improving machining efficiency and ensuring hole quality [11,12]. Schützer et al. [13] studied the impact of manufacturing error and the geometry of cutters on the quality of holes when machining sintered steel alloy materials, and found that expanding the surface of contact between the secondary cutting edges and hole wall surface could improve the straightness tolerance class. By analyzing the cutting force, shape error and surface finish of the machined surface [14], a reaming process function was proposed which found that suitable cutting parameters could improve the standard deviation and accuracy grade. Li et al. [15] studied the geometric error and surface quality after reaming the bronze–aluminum-alloy-stepped holes, which showed that PCD tools have better geometric accuracy error and diameter steadiness when machining the beryllium–copper alloy.

From the above research on hole machining, it can be found that there are many studies on the interactions among tool wear, hole quality, and cutting parameters during machining, while the studies on machining titanium alloy with carbide reamers are not many. In the present study, a comparative experiment analysis of two kinds of groove reamers is presented. The effects of the right-hand reamer and straight groove reamer on the cutting force, tool wear, surface topography, surface roughness, and hole geometric accuracy with different cutting parameters for reaming Ti6Al4V material are studied. Additionally, the other important point is that as the tool diameters of Tool A and Tool B are different, the concepts of relative surface roughness and equivalent cylindricity are put forward, which are transformed into equivalents for comparison.

## 2. Materials and Methods

### 2.1. Experimental Equipment

The experiment was carried out in a four-axis horizontal machining center DMG MORI-NV5000 α1 HSC (GILDEMEISTER, Bielefeld, Germany). The workpiece material was held on a fixture that was fixed on the panel of the dynamometer by bolting. The Kistler 9255C dynamometer device (Sindelfingen, Germany) was selected to measure the cutting force. The force signal was collected and processed by the data-acquisition system Kistler 5435, and then amplified by the charge signal amplifier Kistler 5067. The changing cutting force signal was displayed by the software Dynoware (3.0.9.0). The reamer was clamped in an HSK holder, and its run out was 2 μm. Water-soluble coolant was externally applied to the tool with a pressure of 3.5 MPa. The equipment of the reaming experiment is exhibited in Figure 1.

### 2.2. Tools and Workpiece Material

The tools used in the experiment were the right-hand and straight groove integral cemented-carbide reamers, which were manufactured by Guohong Tools System (Wuxi) Co., Ltd., Wuxi, China. The matrix was high-cobalt cemented carbide composed of a 12% Co binder with a WC grain size of 0.6~0.8 μm. TiAlN was adopted for the coating and the thickness was 2~3 μm, which had excellent critical wear resistance. The right-hand reamer was marked as Tool A, and the straight groove reamer was marked as Tool B, as shown in Figure 2. The detailed dimensional information of Tool A and Tool B is shown in Table 1.

The workpiece was Ti6Al4V material with a plate of 70 × 70 × 15 mm, and its chemical element quality composition is shown in Table 2. To ensure flatness, the surface was first machined by fine milling. The diameter of the pre-drilled holes was 13.8 mm and 15.8 mm, respectively, and the depth of the cut remained an unchanging value of 0.1 mm.

### 2.3. Testing Methods

So as to further study the effect of the tool structure and machining parameters on cutting performance, the experiments adopted a full factor method which is displayed in Table 3. The machining parameters were selected according to the references [17,18,19] and the actual production machining experience of the factory.

The microscope KEYENCE VHX-6000 (Osaka, Japan) and Evo 18 high-resolution field emission Scanning Electron Microscope were adopted to detect the tool wear forms. In order to facilitate the detection and analysis of the hole wall surface, the wire-cutting machine tool was used to conduct axial split sampling at the position parallel to the axis as close to the hole wall as possible. The diameter and cylindricity were measured by three Coordinated Measuring Machines. The surface roughness was measured with Mitutoyo SJ210 (Kawasaki, Japan). Three surfaces were randomly selected for measurement at each hole, and the average values of the surface roughness and cylindricity were recorded and calculated.

## 3. Results and Discussion

### 3.1. Cutting Force

The cutting force signal collected in the reaming process is filtered, and the average cutting force value of the stable cutting cycle is selected as the reaming force for analysis. The force at a specific measurement time marked as tangential force ($F_{xi}$), feeding force ($F_{yi}$), and thrust force ($F_{zi}$) are logged. Formula (1) is used to calculate the average cutting forces in the *x*, *y* and *z* directions, as shown below.
(1)$$F_{x}=\frac{1}{n}(\sum_{1}^{n}F_{xi}),\;F_{y}=\frac{1}{n}(\sum_{1}^{n}F_{yi}),\;F_{z}=\frac{1}{n}(\sum_{1}^{n}F_{zi})$$
where $F_{xi}$, $F_{yi}$ and $F_{zi}$ are each point of the reaming force in the direction of *x*, *y* and *z*, respectively. *n* is the sum of all the selected points. The calculation in Formula (2) is a resultant of the cutting force, as follows [20].
(2)$$F=\sqrt{F_{x}{}^{2}+F_{y}{}^{2}+F_{z}{}^{2}}$$

The fluctuation degree of the cutting force can be regarded as the direct result of cutting stability and surface integrity [21]. Figure 3a shows the curve of the cutting force under different cutting speeds and feed rates with Tool A. It can be seen that when the cutting speed is constant, the cutting force increases with the increasing feed rate. When the feed rate is unchanging, the cutting force first decreases and then increases with the increasing cutting speed. When the feed rate is 0.4 mm/rev, 0.5 mm/rev, and 0.6 mm/rev, the corresponding cutting force changes in the range of 35~42 N, 37~45 N and 39~50 N, respectively. When the cutting speed is 40 m/min, the curve of the cutting force appears as an inflection point, and the minimum value of the cutting force is 35 N when the feed rate is 0.4 mm/rev.

Figure 3b shows the change in the cutting force after being reamed with Tool B. It can be seen that the overall cutting force is larger than that of the right-hand reamer Tool A. Similarly, the cutting force of the two kinds of reamers has a similar changing trend with cutting parameters. When the feed rate is 0.4 mm/rev, 0.5 mm/rev, and 0.6 mm/rev, the corresponding cutting force ranges from 61 to 73 N, 70 to 80 N, and 79 to 88 N, which is an increase of 79%, 87% and 84% compared to Tool A, respectively.

The reason that the cutting force increases with the feeding rate is that the increasing feed rate raises the cutting thickness of the cutter teeth. When the cutting speed is 30~40 m/min, the cutting force gradually decreases with the increasing cutting speed due to the effect of thermal softening. When the cutting speed is 40~60 m/min, the cutting force further increases with the increasing cutting speed [22]. Li et al. [23] pointed out that when the cutting speed is 40~80 m/min, the cutting force increases, that is, the material is mainly plastic deformation, and the strength is enhanced. According to the literature [24,25], the relationship between the cutting force and cutting speed during the machining of Ti6Al4V is also influenced by the interface conditions of the rake face. With the increasing temperature, titanium alloy easily welds itself to the rake face, resulting in the increase in cutting force.

For the right-hand spiral groove reamer, the shear mechanism occurs in the axial and radial directions. On this basis, the dynamics of the cutting structure of the spiral groove reamer occurs in the form of diagonal motion. Due to the vector generated by the two motions, the cutter can help the right-hand spiral groove reamer move axially and radially at the same time, while the cutting mechanism merely occurs in the radial direction of the straight groove reamer. Due to the fact that the spiral angle and the cutting phenomenon of the spiral groove reamer is similar to drilling, the tool geometry helps to more easily discharge a large amount of chips from the cutting area, so as to avoid increasing friction and cutting force [10]. Therefore, the cutting force generated by the right-hand spiral groove reamer Tool A is smaller than that of the straight groove reamer Tool B.

### 3.2. Tool Wear

It can be seen from Figure 4a that there is adhered material and a formed staired face on the rake face of Tool A. Coating peeling off is observed on the rake face shown in the partially enlarged view in Figure 4 P_1_, and on the cutting edge, micro-chipping is observed in Figure 4 P_2_. It can be seen from Figure 4b that there is a built-up edge (BUE) on the cutting edge of Tool B. Generally speaking, a BUE appears at low speeds during cutting ductile materials. Due to the low cutting speed in this study, a BUE is inevitable during the machining of Ti6Al4V materials [26]. As is known, the cutting zone produces higher temperature owing to the low thermal conductivity of the Ti6Al4V material and the work-hardening effect. The material adhesion and the BUE occur on account of the high temperature and high pressure during machining and the high chemical activity of the cemented-carbide reamer to the Ti6Al4V material. When the attached material falls off during machining, the new cemented-carbide matrix is easily exposed, which reduces the tool strength and leads to fast tool wear failure [27].

Figure 5 shows the tool wear morphologies of the rake face and margin of Tool A with the KEYENCE microscope. It can be seen from Figure 5a that there is micro-chipping and abrasive wear on the rake face. There is micro-chipping and notch wear on the margin shown in Figure 5b, which would lead to the exacerbation of the machined surface quality. The margin wear results from the friction between the freshly machined surface and the tool contact region. As is known, notch wear is the result of the combined action of oxidation wear and adhesive wear mechanisms [28]. The reason for the micro-chipping is that the cutting force on the edge surpasses the collapsing strength of the tool material and causes microcracks. Along with cutting, micro-chipping occurs. In addition, cemented carbide is a brittle material, and brittle edge chipping is very common in cutting titanium alloy [8].

As shown in Figure 6a, Tool B is seriously damaged at the cutting edge. Breakage failure and crater wear could be observed on the rake face edge. The main wear mechanism of the cemented-carbide tool in the machining of aeronautical titanium alloy parts is crater wear [29], and the crater wear mechanism has been widely accepted in the machining of titanium alloys [24,30]. When cutting plastic materials, if the cutting speed and thickness are large, then a crescent depression appears on the rake face. The position of the crater wear occurs at the region with the highest cutting temperature on the tool rake face. In the process of wear, the width and depth of the crater wear increase continuously. When the crater wear extends to the narrow edges, the strength of the cutting edge is greatly weakened, which easily causes edge collapse.

Compared with other cutting edges, the damage failure in Figure 6a is the most serious. It can be inferred from the serious failure morphology that some workpiece materials are bonded on the cutting edge. A high shear stress is generated between the bonding layer and the tool matrix, resulting in part of the material on the rake face being sheared and torn off. The chemical mutual effect between the tool and the workpiece is conducive to crater wear [31]. There is also micro-chipping at the margin shown in Figure 6b, which is similar to Tool A.

From the comparison and analysis of the above tool wear forms, it can be inferred that the right-hand reamer Tool A is slightly worn, with only mild edge collapse, material adhesion and coating falling off, while the straight groove reamer Tool B has serious damage failure. The main reason is that the straight groove reamer is subject to a large force in the cutting process, which easily leads to tool damage failure.

After Tool A and Tool B machine the same number of holes, respectively, the wear band width on the rake face of Tool A is measured to be 59 μm in Figure 5a, the width of the wear band at the margin is measured to be 24 μm in Figure 5b, while the width of wear band of Tool B is 106 μm in Figure 6a. Compared with Tool A, the width of the wear band increases by 80%. The width of the wear band at the margin is 34 μm in Figure 6b. Compared with Tool A, it increases by 42%. Therefore, after machining a certain number of holes, Tool B reaches the standard service life first. Therefore, the service life of the right-hand reamer Tool A is better than that of the straight groove reamer Tool B.

### 3.3. Surface Quality

Surface quality is one of the main standards to evaluate the accuracy of processing. Surface defects deeply influence the ability of parts to bear yield stress, high temperature, and wear, leading to surface cracking and transformation, thus affecting the service life and dependability [32,33].

#### 3.3.1. Surface Topography

Surface topography is one of the main features used to evaluate machining quality. Figure 7 and Figure 8 show the surface topography reamed by Tool A and Tool B, respectively, at different machining parameters. From the surface topography of the hole wall, it can be seen that obvious feed marks appear with the increasing feeding rate; the surface quality of the hole gradually becomes worse. Some small protrusions are observed, as shown in Figure 7c, owing to the poor surface quality of the initial hole [14]. The chips produced during reaming could not be discharged in time, and the tool easily vibrates, resulting in feed marks on the hole wall surface. Pits of different sizes appear on the surface. There is strong extrusion and friction between the margin and the machined hole wall surface. With the action of high temperature and pressure, the cemented-carbide reamer and the Ti6Al4V material chip debris have an interaction with each other, bonded to the rake face, which scratches the newly machined hole wall surface and leads to serious notch wear, as shown in Figure 8b,c. If adhesion wear and notch wear are observed at some positions on the surface, then it is unacceptable for parts in the aerospace field because it would cover delamination and other surface flaws, leading to the premature service life of the product parts.

It can be seen from the topography comparisons that the surface quality machined by the right-hand reamer Tool A is better than that machined by the straight groove reamer Tool B. There are only a few feed marks and protrusions on the surface machined by Tool A, while there are a lot of feed marks, debris adhesion, and notch and pit wear on the surface machined by Tool B. The main cause is that Tool A is right-handed. The chips are easily discharged upward with the spiral groove and the cutting force is small, so the tool does not easily vibrate, which is conducive to the improvement of the machining surface quality. Tool B has a straight groove, so the chips are not easily discharged and are easily blocked in the hole, scratching the surface and resulting in poor surface quality. Moreover, the straight groove reamer easily vibrates and form feed marks due to the large force.

#### 3.3.2. Surface Roughness

Surface roughness is an important indicator of surface integrity. *Ra* is the most important index to evaluate surface roughness. Many studies have been carried out on the theoretical model of surface roughness [34,35,36,37]. Like turning, reaming is a type of continuous cutting, which means that the cutting amount is uniform and the cutting resistance is uniform, and the cutting edge is in contact with the workpiece at all the time during the process of machining, as is shown in Equation (3) [38].
(3)$$Ra=\frac{10^{3}f^{2}}{18\sqrt{3}r}$$
where *r* is the edge radius (mm) and *f* is the feeding rate (mm/r). According to Equation (3), especially the *f* has the greatest impact on *Ra*. Only considering the influence of *f*, the surface roughness analysis results in Figure 9 are in accordance with the result of Equation (3). The feeding rate has the greatest influence on the surface roughness, and the other studies have yielded similar results [39,40,41].

In order to eliminate the influence of diameters of Tool A and Tool B on surface roughness, relative surface roughness is introduced, marked as *M*, and the calculation is as follows:(4)$$M=\frac{Ra}{D_{Tool}}$$
where *Ra* is the surface roughness and $D_{Tool}$ is the tool diameter.

Figure 9a shows the relative surface roughness value obtained by Tool A. It can be seen that the relative surface roughness increases with the increasing feed rate, while it decreases with the increasing cutting speed. When the feed rate is 0.4 mm/rev, 0.5 mm/rev, and 0.6 mm/rev, the value range of the relative roughness is 0.014~0.033, 0.019~0.040 and 0.029~0.054, respectively. Under different combinations of cutting parameters, the relative surface roughness is in the range of 0.014~0.054. When the cutting speed is 60 m/min and the feed rate is 0.4 mm/rev, the minimum relative surface roughness is 0.014.

Figure 9b shows the surface roughness yielded by Tool B, which has a similar changing trend with Figure 9a. Under different combinations of cutting speed and feed rate, the range of the relative surface roughness is 0.018~0.051. When the feed rate is 0.4 mm/rev, 0.5 mm/rev, and 0.6 mm/rev, the variation range of the relative roughness is 0.018~0.033, 0.022~0.037 and 0.029~0.051, respectively. When the cutting speed is 60 m/min and the feed rate is 0.4 mm/rev, the minimum relative surface roughness is 0.018.

From the comparison of relative surface roughness in Figure 9a,b, it is found that the relative surface roughness value yielded by the right-hand reamer Tool A is lower at the low feed rate, while the relative roughness value yielded by the straight groove reamer Tool B is low at the high feed rate. For important aerospace parts, the lower the surface roughness value, the better their behavior in service, so a lower feed rate should be selected. In addition, reaming with the right-hand reamer Tool A resembles drilling, and the helix angle contributes to more easily discharging large amounts of chips from the cutting area, avoiding increased friction. Therefore, the helix angle provides better surface roughness, and the same result has been shown in the literature [10].

### 3.4. Geometric Accuracy Error

#### 3.4.1. Hole Diameter

The hole diameter deviation is a measure of dimensional tolerance, which is the difference between the machined diameter and the nominal hole size. The higher the diameter size tolerance class is, the smaller the hole diameter deviation is.

When the cutting speed is 30~60 m/min, it can be seen from Figure 10a that with the increasing cutting speed, the hole diameter deviation gradually increases. When the feed rate is 0.4 mm/rev, 0.5 mm/rev, and 0.6 mm/rev, the variation range of the hole diameter deviation is 3.5~6.0 μm, 2.3~4.5 μm and 1.4~3.0 μm, respectively. With the increasing cutting speed, the temperature in the machining area rises accordingly, resulting in a slight expansion of the material, making the aperture value larger. The diameter decreases with the increasing feed rate because the residence time of the reamer decreases in the hole with the increasing feed rate. The diameter of the right-hand reamer Tool A is Φ14.018 mm, as the H7 tolerance class is 0~18 μm and the worst hole diameter deviation is 6.0 μm. To ensure that the hole diameter is closer to the tool diameter, the parameters should be selected as the cutting speed of 40 m/min and the feed rate of 0.6 mm/rev.

It can be seen from Figure 10b that the hole diameter deviation decreases first and then increases with the increasing cutting speed. When the feed rate is 0.4 mm/rev, 0.5 mm/rev and 0.6 mm/rev, the variation range of hole diameter deviation is 8.8~16.8 μm, 7.4~15.2 μm and 6.5~15 μm, respectively. The hole diameter deviation of the machined hole reamed by Tool B is also 0~18 μm due to the diameter of Tool B being 16.018 mm. It can be seen that the hole diameter deviation is within the H7 tolerance class, while the hole diameter deviation is larger compared with Tool A. In order to ensure that the hole diameter is closest to the cutting tool, the machining parameters should be selected as the cutting speed of 50 m/min and the feed rate of 0.6 mm/rev.

The hole diameter is slightly larger than the diameter of Tool A and Tool B, respectively, which might be primarily due to the light deviation of the positioning accuracy of the reamer during the machining process and the circumferential run out and inclination of the spindle [14]. The run out of the spindle is 2 μm in this study. There are other factors that lead to diameter errors, such as the unstable radial direction of the reamer during feeding, the excessive feed rate, the factors of the coolant, especially the serious tool wear, etc. The heat produced in the cutting process results in the thermal expansion of the tool and workpiece, thus affecting the diameter of the hole [42]. It is found that the right-hand reamer Tool A has a lower hole diameter deviation when machining titanium alloys with various parameters compared to the straight groove reamer Tool B.

#### 3.4.2. Cylindricity

Cylindricity refers to the value between the utmost and the minimal size of the vertical section at any position of the hole, that is, the deviation degree of the whole cylinder, as shown in Figure 11.

In order to eliminate that the influence of tool diameter, which might be responsible for cylindricity, the “Equivalent cylindricity” is introduced, marked as *Q*, and the calculation is as follows:(5)$$Q=\frac{P}{D_{Tool}}$$
where *P* is the original cylindricity and $D_{Tool}$ is the tool diameter.

Figure 12a shows the equivalent cylindricity of the hole reamed by Tool A. It can be seen that with the increasing cutting speed, the equivalent cylindricity first decreases and then increases. When the cutting speed is constant, the equivalent cylindricity increases with the increasing feed rate. When the cutting speed is 30 m/min, 40 m/min, 50 m/min and 60 m/min, the variation range of the equivalent cylindricity is 0.271~0.493, 0.157~0.300, 0.293~0.479 and 0.321~0.557, respectively. The equivalent cylindricity achieves the minimum value of 0.157 when the cutting speed is 40 m/min and the feed rate is 0.4 mm/rev.

Figure 12b shows the cylindricity of the hole made by Tool B. It can be seen that the equivalent cylindricity has a similar changing trend with Figure 12a. When the cutting speed is 30 m/min, 40 m/min, 50 m/min and 60 m/min, the equivalent cylindricity ranges from 0.325 to 0.456, 0.240 to 0.360, 0.310 to 0.440 and 0.470 to 0.560, respectively. When the cutting speed is 40 m/min and the feed rate is 0.4 mm/rev, the equivalent cylindricity achieves the minimum value of 0.240. From the comparison between the equivalent cylindricity values of Figure 12a,b, it can be seen that the equivalent cylindricity yielded by the right-hand reamer Tool A is generally lower than that of the straight groove reamer Tool B, that is, the holes machined by Tool A show better cylindricity than Tool B.

The results of the cylindricity show that when the cutting speed is 30~40 m/min, the accuracy of the equivalent cylindricity increases with the increasing cutting speed. When the cutting speed further increases, the cutting vibration also increases as the tool has little contact with the hole wall along the hole axis, and the following effect on the hole is not obvious when cutting in and out, leading to poor cylindricity and easily causing an uneven top and bottom. Therefore, the proper selection of the cutting parameters is very important to improving the cylindricity accuracy. The combination of the parameters of a cutting speed of 40 m/min and a feed rate of 0.4 mm/rev is the most suitable.

## 4. Conclusions

The purpose of this study is to investigate the machining performance of Ti6Al4V material reamed by cemented-carbide reamers with two kinds of groove structures under different cutting parameters. Some conclusions can be drawn as follows:(1)Due to the groove structure, the shear mechanism occurs in the axial and radial direction for the right-hand reamer Tool A, while it merely occurs in the radial direction for the straight groove reamer Tool B, leading to the cutting force of the right-hand reamer Tool A being smaller than that of the straight groove reamer Tool B.(2)Micro-chipping and material adhesion appear on the rake face of the right-hand reamer Tool A, while serious damage failure occurs on one cutting edge of the straight groove reamer Tool B. The wear of the margin plays an important role in determining the quality of the machined surface.(3)The topography is better machined by the right-hand reamer Tool A compared to the straight groove reamer Tool B. In order to obtain lower surface roughness, a combination of a cutting speed of 60 m/min and a feed rate of 0.4 mm/rev should be selected.(4)With different cutting parameters, the holes machined by the right-hand reamer Tool A have a lower hole diameter deviation. The combination of a cutting speed of 40 m/min and a feed rate 0.4 mm/rev could achieve a high geometric accuracy of cylindricity, and the holes machined by the right-hand reamer Tool A show better cylindricity.

## Figures and Tables

**Figure 1 materials-15-05027-f001:**
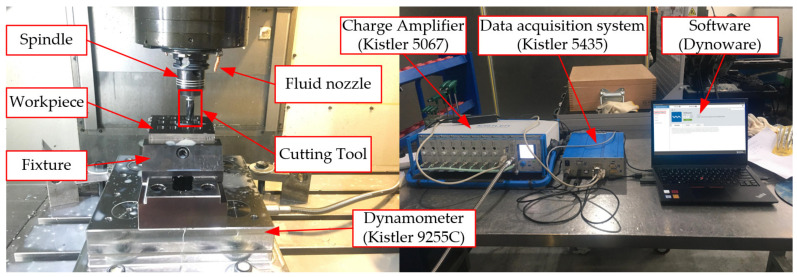
Experimental equipment.

**Figure 2 materials-15-05027-f002:**
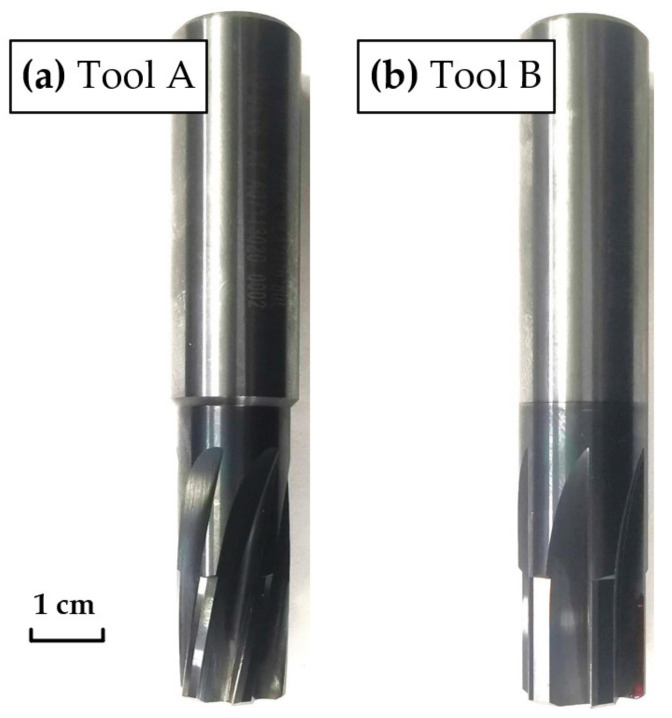
Cutting tools: (**a**) Right-hand reamer, marked as Tool A; (**b**) Straight groove reamer, marked as Tool B.

**Figure 3 materials-15-05027-f003:**
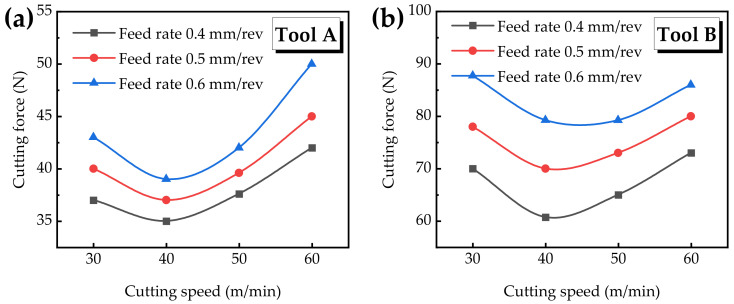
Cutting force versus different cutting parameters.

**Figure 4 materials-15-05027-f004:**
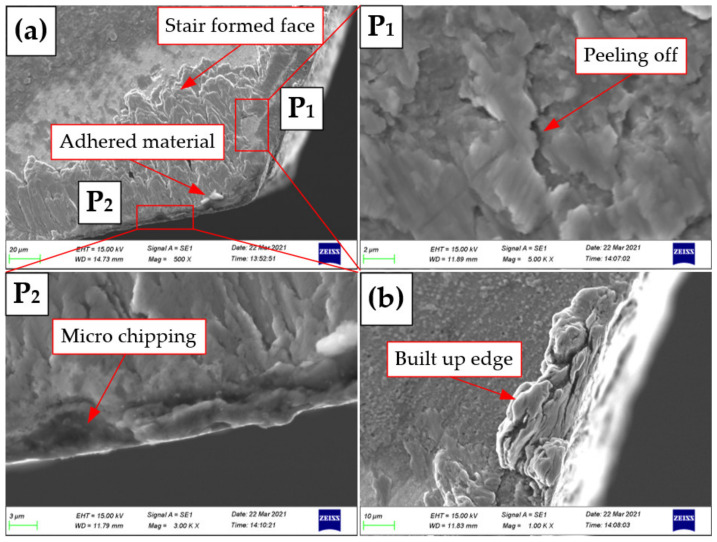
Tool wear morphologies (SEM): (**a**) Tool A, **P_1_** and **P_2_** are partially enlarged views of the rake face on Tool A; (**b**) Tool B.

**Figure 5 materials-15-05027-f005:**
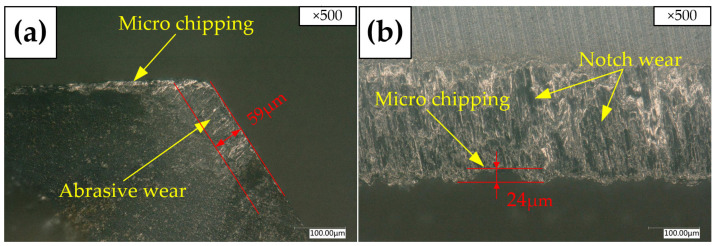
Tool wear morphologies with KEYENCE microscope of Tool A: (**a**) Rake face; (**b**) Margin.

**Figure 6 materials-15-05027-f006:**
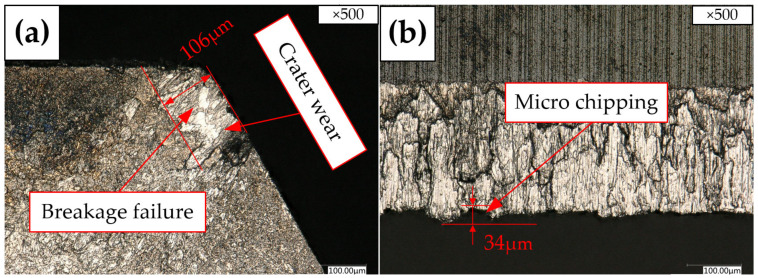
Tool wear morphologies with KEYENCE microscope of Tool B: (**a**) Rake face; (**b**) Margin.

**Figure 7 materials-15-05027-f007:**
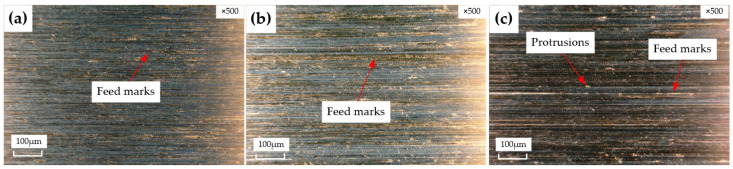
Surface morphologies reamed by Tool A with KEYENCE microscope, (**a**): v=60 m/min, f=0.4 mm/rev; (**b**): v=60 m/min, f=0.5 mm/rev; (**c**): v=60 m/min, f=0.6 mm/rev.

**Figure 8 materials-15-05027-f008:**
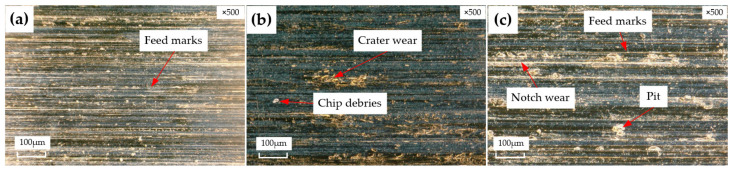
Surface morphologies reamed by Tool B with KEYENCE microscope, (**a**): v=60 m/min, f=0.4 mm/rev; (**b**): v=60 m/min, f=0.5 mm/rev; (**c**): v=60 m/min, f=0.6 mm/rev.

**Figure 9 materials-15-05027-f009:**
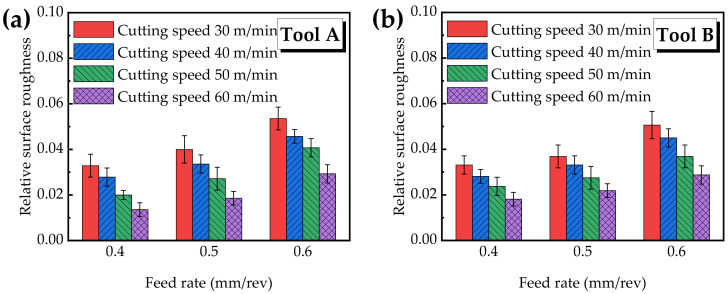
Relative surface roughness versus different cutting parameters.

**Figure 10 materials-15-05027-f010:**
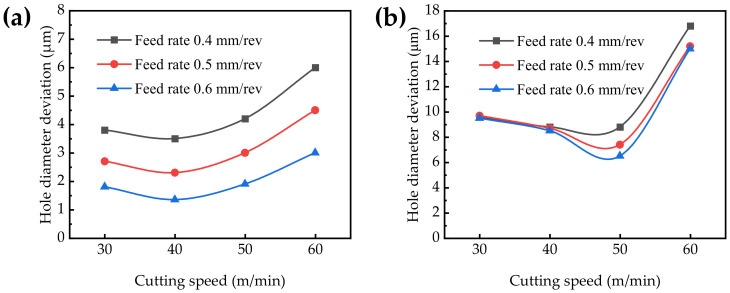
Hole diameter deviation versus different cutting parameters: (**a**) Tool A; (**b**) Tool B.

**Figure 11 materials-15-05027-f011:**
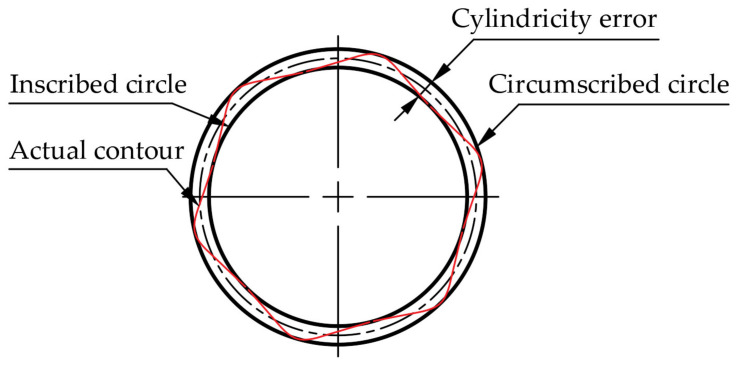
Schematic diagram of cylindricity cross section.

**Figure 12 materials-15-05027-f012:**
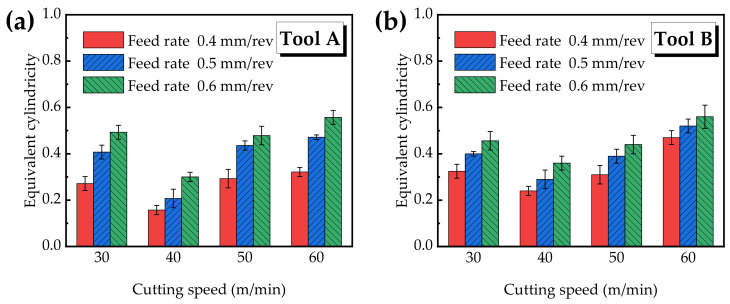
Equivalent cylindricity versus different cutting parameters.

**Table 1 materials-15-05027-t001:** The dimensional information of Tool A and Tool B.

Item	Tool A	Tool B
Diameter	14.018 mm	16.018 mm
Length of cut	15 mm	15 mm
Length of flute	30 mm	30 mm
Direction of flute	Right	Straight
Number of flutes	6	6
Rake angle	8°	8°
Helix angle	12°	0°
First clearance angle	10°	10°
Second clearance angle	25°	25°
Margin	0.2 mm	0.2 mm
Clamping diameter	16 mm	16 mm
Length overall	80 mm	80 mm

**Table 2 materials-15-05027-t002:** Chemical element quality composition of the workpiece material Ti6Al4V [16].

Element	Ti	Al	V	Fe	C	N	H	O
wt%	Base	5.5~6.75	3.5~4.5	<0.25	<0.08	<0.05	<0.01	<0.2

**Table 3 materials-15-05027-t003:** Full factor experimental design: the machining parameters of Tool A (#1~#12) and Tool B (#1~#12) in reaming Ti6Al4V.

Number	Feed Rate (mm/rev)	Cutting Speed (m/min)
#1	0.4	30
#2	0.4	40
#3	0.4	50
#4	0.4	60
#5	0.5	30
#6	0.5	40
#7	0.5	50
#8	0.5	60
#9	0.6	30
#10	0.6	40
#11	0.6	50
#12	0.6	60

## Data Availability

Not applicable.

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
