# Peer review of "Study on Characteristics for Reaming Titanium Alloy Ti6Al4V with Two Kinds of Cemented-Carbide Groove Reamers"

_materials, 2022, doi:10.3390/ma15145027_

Round 1
Reviewer 1 Report
The manuscript titled “Study on Characteristics for Reaming Titanium Alloy Ti6Al4V 2 with Two Kinds of Groove Cemented Carbide Reamers” presents the comprehensive investigation of the different kinds of groove reamers, studied on Ti6Al4V alloy material, and analysis based on the determined effects of right-hand reamer and straight groove reamer on cutting force, surface quality, tool wear, and hole geometric accuracy, with different cutting parameters. The main contributions of this paper is determining the machining performance of Ti6Al4V alloy following the reaming with two kinds of groove reamers, and the understanding of the adequate cutting force mechanism and effects on the machined holes.
Reviewer 2 Report
This paper delas with an experimental investigation on performance of two different types of reamers while machining Ti-6Al-4V. Authors have selected right hand reamer and straight groove reamer for the comparative study and experiments were conducted at different feed and cutting speed. Machining performance was assessed based on the cutting force, tool wear, surface quality and geometric accuracy error. It was concluded that right hand reamer gives better performance. Before accepting this manuscript for publication it needs to be thoroughly revised by incorporating following points.
1. Novelty of this work needs to be stated clearly apart from workpiece and tool combination.
2. In this study authors have selected one tool with right hand reamer and another as straight groove reamer. It will be an obvious result that, right hand reamer will perform better compared to straight groove reamer. Rather authors should have selected different types of right-hand reamer with different helix angles or rake angle, etc. and then conduct a comparative study.
3. It would have been better show the images of Tool A and Tool B in section 2.2
4. What was the basis for selecting cutting parameters as specified in Table 3
5. Page No: 5 – Figure 3 shows various types of tool wear observed in this study. It would be better to include EDS analysis to confirm coating peel off, micro chipping, adhesion of material and built-up edge formation
6. Section 3.2 - As the formation of BUE is detected in this study, chances of crater wear is high. Authors did not mention anything about the crater wear in this study.
7. It would be better to include progress of flank wear in this study. In terms of tool life, how far Tool A is better than Tool B?
8. Section 3.3.2, Page No: 7 - Equation (1) is for turning operation. How it can be used for reaming operation? Justify this.
Reviewer 3 Report
The study compared two tools with similar dimensional parameters, except three are different (Diameter, Helix angle, Clamping diameter) as in table 1. The machining parameters ( Feed rate and Cutting speed ) are varied in the 12 experiments, as in table 3.
1- The study did not distinguish which of the variable dimensional parameters has the role of increasing or decreasing the cutting force.
2- Table 3 did not show what type of experimental design was used.
3- The study summarizes that the decrease and increase in the cutting force result from thermal softening and plastic deformation hardening effects without evidence. The process is more complicated than what the study tries to describe in Figure 2.
4- Line 246 (the helix angle provides better surface roughness). What about the tool diameter? It varied also.
5- In the y-axis in figure 9, it is better to use hole diameter tolerance instead of diameter only to compare the two tools.
6- In line 303 (With the same machining parameters, the holes machined by Tool A show better cylindricity than Tool B). Which parameter is responsible for this characteristic, is it the tool diameter or the helix angle, and why?
Reviewer 4 Report
The topic of the article is interesting and also I see the possibility of transferring knowledge directly into industrial practice. The title is adequate to the contribution. The theoretical background is understandable. A content of the main section is presented clearly. The methodology is sufficient. In the experiment suitable measurement technology was used which contributes to the relevance of the data. The processing of results and conclusions is comprehensible and meets the standards of a scientific article. Supporting figures are appropriate. I consider that the article is suitable for publishing in the MDPI Processes, but some details are missing:
- Generally - The authors should present and highlight the scientific novelty. What is new about the results described in the article? Why the research was undertaken and what is the result of it?
(Introduction)
- The introduction should be improved. References should be done in one to one mode for better clarity otherwise it would be just a formality to have literature review without having appropriate focus. Examples:
- line 30 - …. titanium alloys are considered as difficult materials to machine due to their high temperature strength, high chemical activity, low elastic modulus, low thermal conductivity and low deformation coefficient [1-6].
- line 35 - …. to meet dimensional tolerance and surface quality requirements [7-11].
- line 41 - …. important impact on improving tool life and machining quality [12-17].
(Materials and Methods)
- Fig 1 - No markings for dynamometer, amplifier and computer program name.
- No detailed description of the cutting tools materials (symbols, chemical composition). Were the cutting data within the recommended range? No information on depth of cut. Were there pre-holes and what diameter were they?
- Why the components of the forces were not analyzed?
(Results and Discussion)
- The article could present information about a statistical analysis of the research results. For example, are there average values in the graphs (Fig. 2,8,11)? What was the scatter of the results or the standard deviation, etc. Is it possible to present a mathematical formula to determine the optimal values of cutting parameters for reaming Ti6Al4V titanium alloy?
- In the article, there are no results of measurements of parameters describing the wear of cutting tools, eg VB or KT. The authors didn’t present the wear curve of the cutting tool for the analyzed case. What was the tool life? What authors think about the creation of tool wear with the supply of coolant?
Round 2
Reviewer 3 Report
The modified version seems better.